# Microfluidic Lab-on-a-Chip for Studies of Cell Migration under Spatial Confinement

**DOI:** 10.3390/bios12080604

**Published:** 2022-08-05

**Authors:** Federico Sala, Carlotta Ficorella, Roberto Osellame, Josef A. Käs, Rebeca Martínez Vázquez

**Affiliations:** 1Institute for Photonics and Nanotechnologies, CNR, Piazza Leonardo da Vinci 32, 20133 Milano, Italy; 2Peter Debye Institute for Soft Matter Physics, University of Leipzig, 04109 Leipzig, Germany

**Keywords:** lab-on-a-chip, cell migration, microfluidics, PDMS, hydrogels, femtosecond laser microfabrication, two-photon polymerization

## Abstract

Understanding cell migration is a key step in unraveling many physiological phenomena and predicting several pathologies, such as cancer metastasis. In particular, confinement has been proven to be a key factor in the cellular migration strategy choice. As our insight in the field improves, new tools are needed in order to empower biologists’ analysis capabilities. In this framework, microfluidic devices have been used to engineer the mechanical and spatial stimuli and to investigate cellular migration response in a more controlled way. In this work, we will review the existing technologies employed in the realization of microfluidic cellular migration assays, namely the soft lithography of PDMS and hydrogels and femtosecond laser micromachining. We will give an overview of the state of the art of these devices, focusing on the different geometrical configurations that have been exploited to study specific aspects of cellular migration. Our scope is to highlight the advantages and possibilities given by each approach and to envisage the future developments in in vitro migration studies under spatial confinement in microfluidic devices.

## 1. Introduction

Cell migration is a complex phenomenon that plays a key role in a variety of biological processes in multicellular organisms, including embryological development, wound healing and immune response. Pathological processes, such as tumor invasion and metastatic spread, also rely on cell migration [1]. For instance, cells that have evolved into a malignant phenotype can exploit the epithelial-to-mesenchymal transition (EMT) to initiate motion, invade the surrounding tissue and reach distant organs in the body [2,3,4]. In vivo, tissues are generally characterized by a dense three-dimensional scaffolding, the extra-cellular matrix (ECM), consisting of proteoglycans, occupying the extra-cellular space in form of a hydrated gel, and a variety of fibrous proteins, such as collagens, fibronectins, elastins and laminins [5,6]. Given such a complex extra-cellular microenvironment and the high cellular density characterizing biological tissues, in vivo migration and spreading occur under varying degrees of confinement, leading motile cells to constantly adapt their shape and migration strategy.

Confinement is an important mechanical cue for motile cells, as it can trigger a variety of cellular responses [7]. For instance, fibroblasts and mesenchymal tumor cells respond to confinement by switching in between different migration mechanisms in order to successfully navigate through their surrounding ECM. Particularly in the context of tumor metastasis, a matrix of high stiffness and rigidity has been shown to promote tumor cell proliferation [8] and EMT [9], cause multinucleated cell division [10,11], and influence their ability to invade the surrounding tissue and reach distant locations in the body [12]. Similarly, interstitial pressure has been shown to drive malignant cell outgrowth and dissemination throughout the ECM [13,14,15,16]. There are several reviews in the scientific literature regarding studies of cell migration strategies; in particular, we refer the reader to the work of Paul et al. [1] for a deeper insight into it.

Dissemination can occur through single cell invasion or collective invasion, depending on whether the cells have transitioned to a fully mesenchymal phenotype or whether they still retain epithelial characteristics [17,18]. In 3D environments, mesenchymal cells can migrate through the available gaps by using protrusions of different morphology and size driven by actin polymerization such as lamellipodia (thin, fan-like in shape) and filopodia (thin and slender extensions used by the cells to probe the environment) [19,20]. When a cell’s focal adhesions bind to the surrounding ECM, cells can apply traction forces on the matrix and migrate through it using arm-like projections known as pseudopodia [21]. When the activity of cell adhesions is inhibited or suppressed, cancer cells can switch their migration strategy to an amoeboid-based motility mode, thus acquiring a rounded morphology, diffuse distribution of adhesion proteins and spherical, actin-free protrusions (membrane blebs) to drive cell locomotion [22,23,24,25,26], as observed also in in vivo assays [25].

When escaping the primary tumor and moving through the ECM, tumor cells squeeze through 1–30 μm wide pores [27,28] and 1–2 μm wide openings between endothelial cells to access the bloodstream. Once they reach the bloodstream, cells may experience confinement as they navigate through vessels of 3–4 μm diameter [17,18]. Such confined migration induces significant structural changes in the cell’s cytoskeleton [29,30] and focal adhesions while also affecting the nuclear shape and the distribution of Lamin A/C (a protein present in the nuclear lamina whose role is to protect the nuclear envelope from external stress and potential rupture) [31,32,33,34,35,36].

Because of its clinical relevance particularly in the context of cancer research, understanding the effects that confinement and related mechanical cues such as stiffness and shear stress can have on cellular behaviors may prove to be a valuable contribution to the development of strategies and therapies tackling metastatic spread at its early stages. Given its importance, many in vitro systems have been developed in order to study and characterize cellular motility and migration patterns as well as to replicate the physiology of human tissues and in particular ECM mechanical, structural and chemical characteristics.

The four main standard techniques used to study cell migration, whether in 2D or in 3D environments, are the scratch assay, the cell exclusion zone assay, the Boyden chamber and patterned lines [37]. In a scratch assay, which was initially used to study the wound healing for epithelial and mesenchymal cells [38], the target cells are seeded on a Petri dish or in a well plate. Once they attach to the surface and form a uniform monolayer, cells are removed from a certain discrete area (scratching) with the help of a pin tool or a needle. By observing the subsequent migration of the cells and acquiring several images of the edges of the “artificial wound”, it is possible to infer the characteristics of their motility behavior. For instance, multiple assays can be parallelized in a multi-well plate, adding different molecules on top of the scratch to study possible inhibition or promotion effects on the cell migration. Moreover, the assay plate can be covered with an ECM in order to characterize its interaction with the migrating cells. This test is simple and straightforward, and the time-lapse images acquired during the experiment can give important information on cell morphology during migration. However, it lacks reproducibility and standardization, as it is hard to guarantee that the monolayer condition and the scratching procedure are exactly the same within different experiments. Furthermore, this mechanical action can damage the deposited ECM, falsifying the results. During the years, several options have been presented and commercialized in order to create a reproducible scratching process, such as laser ablation or electrical wound, but these dedicated systems increase the complexity of the assay.

A similar assay that aims to avoid the variability in the scratch process is the cell exclusion zone assay [39]. In this test, a mechanical barrier is placed in contact with the cell culture plate before the seeding. The target cells are then cultured around the barrier as in the scratch assay until they reach the uniform monolayer structure, and only then the barrier is removed. The collective migration of the cells toward the newly exposed surface is then studied as in the previous case. This second assay has the advantage of avoiding the scratching procedure, preventing possible damages to the ECM. Furthermore, the wounds created in this way are reproducible in dimension.

The Boyden chamber [40] is a system for transmembrane migration assay. It consists of two chambers separated by a porous membrane. The target cell population is seeded in the first chamber, while the solution to be tested is placed in the second one, generating in this way a chemical gradient and inducing a chemical signal-driven cell migration (chemotaxis). After a given time, the membrane is fixed and stained, and the cells on the second chamber side of the membrane are counted using microscope observation. This system has been widely used and commercialized also in combination with multi-well plates in order to increase the parallelization level and the throughput of the analysis. The main advantage of this system is the possibility to test both adherent and non-adherent cells. Furthermore, the membrane can be coated with ECM proteins to mimic different migration conditions. Moreover, choosing membranes with different average porous sizes makes it possible to investigate the effect of cell spatial confinement on their three-dimensional migration properties, which is in contrast with previously described assays that only allow the investigation of 2D unconstrained motion. However, by using this system, it is not possible to observe the cells during their migration and, in case of too few cells counted out of the membrane, statistical analysis might not be applicable.

The last technique standardly used to study cell migration is the so-called “patterned lines”. In this case, a planar substrate is patterned with proteins that can drive cell adhesion, so that the seeded cells naturally stretch along the defined geometry (for instance, lines). These long lines allow the study of cell migration, imposing a lateral constrain induced by the limited contact area. This technique allows the engineering of the spatial stimuli by changing the shape of the patterned lines, but it does not give hints of the behaviors of cells under mechanical confinement, as in the case of migration through pores.

The previously described assays constitute the pillar of cell migration studies, but they have some limits, i.e., they are intrinsically two-dimensional (2D) techniques, they do not allow a perfect control of the biological conditions, and they have a poor reproducibility. In this framework, a technological improvement to overcome these critical points is needed. Microfluidics has proved to be a new enabling technology for biological assays and in particular in the case of cell migration. One first strong advantage of microfluidics is the precise and reproducible control on the biological environment, i.e., the possibility to induce precise flow rate, control the temperature and the composition of the buffer and even the ability to induce chemical gradients with different shapes thanks to engineered micromixer geometries [41]. Secondly, the system can be optionally embedded with sensors in order to automatically count the migrating cells or to measure their characteristics [42,43]. Third, advanced fluidic geometries can be used to manipulate the cells in a gentle way [44] or to provide external stimuli on demand. Lastly, thanks to the modern microfabrication techniques, it is possible to shape with micrometric precision level the migration area, either in 2D or in 3D. This latter possibility is particularly interesting when studying cell migration under physical confinement, as it allows the engineering of the spatial stimuli of the ECM, changing the environment geometry [45]. The main focus in this field of research is, nowadays, to realize customized systems, tailored on the specific need of the biologists, in order to empower their analysis capability, standardize the experimental procedure and investigate characteristics of the cells that could not be distinguished with standard techniques.

In recent years, many different microfluidic platforms have been proposed to study migration, under spatial confinement, of adherent cells. Their characteristics depend mainly on the material and fabrication method that set important aspects as the minimum constrain dimensions or the 3D level. In the present work, we present a review of different approaches of microfluidic cell migration assays (MCMAs) under spatial confinement. We discuss their technological aspects such as fabrication technique, material or design.

As we will show, despite their designs or dimensions, or the biological problem they aim to study, it is possible to classify them depending on the material and/or the fabrication technique. In this framework, we identify three main categories for the existing MCMAs platforms: namely, those made in polydimethylsiloxane (PDMS), in hydrogels, and those made by ultrafast laser writing in glass and/or photopolymers (see Table 1). The possibility for the cells to adhere to the substrate can strongly influence the choice of migration strategy, co-participating with the spatial confinement [46] to the ECM influence on cellular behavior. For this reason, most of the MCMAs are performed in coated devices (as reported in Table 1). For each category, we will also present the different microfluidic chamber designs and highlight their potentialities in terms of design complexity and geometrical limits.

## 2. PDMS-Based Microfluidic Devices

PDMS is a broadly used material in microfluidics [75,76,77]. It is an inexpensive, transparent and flexible silicon-based polymer and it is in general inert, non-toxic and non-flammable. It is widely used in replica molding and soft lithography, which are processes that can be carried out by non-specialized personnel. The standard fabrication protocol for the realization of PDMS microfluidic devices is divided in three steps, which are schematically reported in Figure 1: the realization of the mold, the replica molding in PDMS and the sealing of the patterned PDMS. In the first step, the microfluidic network is designed and realized with standard microfabrication techniques as a negative copy. After that, the liquid PDMS is poured on the mold and is cured at a temperature of 60–80 ${}^{\circ }$C. Afterwards, the PDMS membrane is peeled off from the mold, thus providing open channels on the surface. The microchannels are then sealed by sticking the PDMS membrane on the top of a flat surface. Following this type of procedure, the microchannels are arranged in an intrinsically planar configuration, but 3D fluidic networks can be obtained using a multilayer configuration or sacrificial layer approach [78,79]. It is important to notice that after the sealing on a transparent substrate, it is possible to optically access the migration area with a standard objective, allowing live imaging of the cells during their migration.

Exploiting standard soft lithography, it is possible to easily reproduce arbitrary 2D geometries with a minimum feature size of few micrometers. To further reduce the channel dimensions down to the nanometer scale, more advanced techniques can be used, such as controlled collapsing of microchannel structures [80] or induced nanocrack formation [81,82].

In addition to being very easy to manipulate, PDMS has some known characteristics that must be taken into account during microfluidic device design [83]. PDMS is not rigid and has a higher stiffness than biological tissues (1–3 MPa), thus inducing undesired fluidic capacitance and presenting migration media mechanical properties that are different with respect to in vivo experiments. As a second aspect, PDMS is permeable to gas exchanges, such as oxygen. This property might be exploited to integrate it in culture chambers, guaranteeing good oxygenation of the sample, but at the same time, it can affect the experiment conditions, for instance in the case of undesired medium evaporation through the material. Another important issue is linked to the absorption of small molecules by the elastomer. This is well known in the field of drugs tests and cellular signaling, as it can act as an important and impeding bias in these kinds of assays. In the same way as PDMS can absorb molecules, it can also release them in the form of un-cross-linked compounds, with consequent cytotoxicity. Some solutions have been suggested to reduce these issues, such as coating with wax paraffine or parylene [84].

In the field of the study of confined cell migration, it is possible to exploit different geometries, with increasing complexity. In the following, we will present the main strategies in order to give an overview of the implementation of microfluidic channels for migration assays, with a focus on the possibility to tailor the geometries addressing specific biological aspects.

### 2.1. Arrays of Straight Channel as Cell Migration Assays

The most simple geometry of a PDMS biochip used to study cell migration and chemotaxis consists of an array of straight channels connecting two reservoirs. As in Boyden chambers, the first reservoir is used for cell seeding, while the second contains the solution that is used to create a chemical gradient along the microchannels. Differently from Boyden chambers, using these MCMAs, it is easy to observe the cells migrating in the channel and study their motion by putting the PDMS biochip on a standard microscope. A simple configuration of this type is presented by Rolli et al. [47], where the two reservoirs were connected by multiple parallel channels in order to increase the throughput of the assay. They studied the migration velocity of human pancreatic epithelial cancer cells (Panc-1) both inside the microchannels and using a patterned fibronectin lines on a culture dish, comparing the migration strategies with and without mechanical confinement. Their results show how the motion of the cells inside the microchannels is faster and more similar to a constant sliding compared to a push-and-pull motion observed for unconstrained cell migration. If the dimension of the channel is comparable to or smaller than the average cell diameter, the device can be used to simulate mechanical confinement. Using this type of configuration, it is possible to study which are the minimum dimensions of the channel cross-section that can be engaged by the cells during the migration and whether the increased contact surface, with respect to the 2D planar motion case, can influence the cell migration strategy. An example of this type of experiment is reported in Figure 2a. Using this geometry, Fu et al. [48] compared the migration speed of two different human breast cancer cell lines (MDA-MB-231, highly metastatic, and MCF7, poorly metastatic), depending on the different cross-sections of the microchannels (fixed height of 5 μm and variable width from 4 to 12 μm). Taking advantage of the optical accessibility of the microfluidic platform, they were able to observe the deformation of the stained nuclei of the cells during the migration. As a second step of the experiment, they added a protein (59-deoxy-59-methylthioadenosine) to inhibit chromatin condensation and reduce nuclear deformability, and they measured a reduction in cell invasiveness, depending on the protein concentration. On the other hand, Tong et al. [49] designed a microfluidic device where the chemical gradient in the second reservoir is controlled and kept constant thanks to a hydrofluidic focusing approach, resulting in a stable concentration profile used to study multiple cell lines, including human osteosarcoma cells (HOS), human breast adenocarcinoma cells (MCF-7, MDA-MB-231) and non-tumorigenic mammary epithelial cells (MCF-10A). In order to study in detail the effect of the chemical gradient combined with mechanical confinement, Irimia et al. [50] realized a MCMA with a more complex geometry, as reported in Figure 2b. The device includes a central area where two parallel channels are connected by an array of straight constrictions, as in the previously described chips. Before this region, the two channels are connected so that the two fluxes of the chemoactractant and buffer are in contact (see bottom part in the figure, labels A and B). The two fluxes do not merge thanks to the laminar regime, but rather, thanks to the contact zone, any possible pressure difference between the two inlets is equalized at the interface. In this way, it is possible to maintain a linear chemical gradient across the migration area despite possible experimental errors in the fluxes calibration, obtaining a robust and reliable device. They also included some microvalves in order to guarantee the proper cell upload in the system, and in this way, they were able to deliver chemical reagents to the front or to the rear of the migrating cell and dynamically study their response. With this device, they studied the migration of leukocytes in a highly confining condition and their response to different drugs: both attractants or inhibitors.

Another important feature that can be added to PDMS-based MCMAs is the possibility to easily coat the capillaries with specific proteins or chemical agents. Straight arrays of capillaries in PDMS have been used as comparative assays in order to explore the effect of the coating on the cells migration, using for instance fibronectin [51] or collagen [52].

The shape of these straight PDMS channels can be engineered in order to induce controlled spatial stimuli on the cells during their migration and study their response. Mak et al. [53] designed a microfluidic system with a series of capillaries that taper from a width of 15 to 4 μm with seven different possible angles in order to investigate the cell invasiveness once facing different spatial constriction gradients. A brightfield microscope image of these channels is reported in Figure 2c. They studied the behavior of bovine aortic endothelial cells (BAECs) as baseline and human breast adenocarcinoma cells (MCF-7, MDA-MB-231), verifying that depending on the tapering angle, these cell lines might invert their migration direction with a higher or lower probability. In the following article [54], the same team investigated the effect of a series of tapered constrictions along the migration channel, as shown in Figure 2d. This PDMS microfluidic device was used to identify different phases of cell migration (MDA-MB-231) once they met a constriction with a dimension smaller than its nucleus size. To be specific, there are four identified phases: “the cell slows down as it reaches the barrier (phase 1), the cell body starts permeating into the barrier (phase 2), the cell pauses or otherwise stops monotonic forward motion (phase 3), and the cell resumes monotonic forward motion and exits the barrier (phase 4)”. Furthermore, they verified that the periodic modulation of the channel dimensions, with a series of constrictions and widenings, can induce morphological changes in invading cells. During the cells’ migration, significantly longer extensions are protruded, and this phenomena may facilitate nutrient finding and chemotaxis homing, and it can be found in metastatic cancer phenotypes.

### 2.2. Advanced Geometries of Fluidic Networks

Up to now, we presented some simple geometries consisting of a series of straight channels with a bi-dimensional geometry, and we introduced possible variations on this scheme. In this section, we will present advanced geometries that exploit the possibility of exploring three-dimensional complexity or integrating several microfluidic components in the same device. Some of these examples are reported in Figure 3.

The first example is a 3D version of the periodic modulation of the straight channel cross-section introduced above. Ma et al. [55] presented an array of 36 straight channels with a periodically modulated height, from 15 to 10 μm. They tested their device with adherent MHCC-97L liver cancer cells and suspended OCI-AML leukaemia cells, finding results analogous and complementary to those presented in [54]: upon periodic mechanical modulation, the cancer cells showed an increased migration velocity as well as an increased plasticity and a cytoskeleton structure alteration.

Raman et al. [56] fabricated a multilayer device with straight channels with a series of bendable PDMS micropillars as the floor. In this way, they forced the cells to migrate on the top of the pillars array, anchoring on their tip, and they studied the direction and the intensity of the traction force depending on the pillar deformation, as shown in Figure 3a. In addition, they included, in the same MCMA, channels with different widths in order to study the difference in migration behavior between a 2D planar case and constricted case.

An additional strategy is to use micropillars to mimic the ECM. In this case, a 2D array of pillar structures with different dimensions and spacing is used to study the migration of cells either atop of it or, in case of migration under constriction, within the array. Doolin and Stroka [58] presented a study on the migration of mesenchymal stem cells in different devices, which were characterized by different pillar spacing, showing different behavior and, for instance, different turning angles during the motion. Furthermore, in the same article, an extended list of works using micropillars-based migration assays is presented. On the other hand, Davidson et al. [59] used pillars with variable diameter and arranged them in a channel-like fashion, as reported in Figure 3b. In this way, they obtained a device varying between a pillar array and a straight channel array in order to simulate the non-continuous spatial constraints found in the natural microenvironment formed by ECM fibers. In particular, these pillar-made channels present constriction points with a minimum dimension of 5 × 5 μm. This porous size is small enough to cause nuclear rupture and strong deformation during the migration, and this feature has been used to study nuclear envelope damage and repair during cancer cell migration [33]. A particular technical aspect of this device was the inclusion of a parallel bypass channel in between the two chambers (seeding and collection) in order to balance the reservoir medium level and guarantee experimental robustness against operator buffer uploading errors.

Boneschansker et al. [57] proposed an advanced geometry to study at the same time the migration of leukocytes toward and away from the chemoattractant, both with or without mechanical constriction. A scheme of the device is reported in Figure 3c. In this device, the chemical gradient is established between two external channels connected by an array of capillaries with small (6–10 μm) and large (50 μm, i.e., not constricting) dimensions. A third channel is located in between the two external channels, splitting the capillaries into left-handed and right-handed halves. This channel is used for the cell upload, and it is equipped with a series of cell traps in order to stop the cells in between the chemical gradient, allowing in this way a migration in both directions. This device was used to study different types of cell lines, such as different neutrophil and leukocytes, and it highlights different migration patterns depending on the combinations of cell type and chemical agent tested.

One last example of high-complexity MCMA is the so-called “fluidic mazes”. A series of microfluidic channels are realized in between the seeding and the collection chamber, including bifurcation, parallel channels with different fluidic resistance and dead-end channels. These complex networks are used to study the “decisions” taken by the cells during the motion and to investigate the mechanism underneath these choices. For instance, it is possible to study the response of the cells to different degrees of fluidic resistance, to different mechanical stimuli [61] or the feedback mechanism based on a self-induced chemical gradient, which is used to navigate the cells away from dead-end structures [60]. These last studies are extremely interesting and can help scientists investigate the behavior of cell population migration, moving the focus of the assay from the single-cell motion mechanism to collective strategies.

## 3. Hydrogels-Based Microfluidic Devices

Hydrogels are a class of three-dimensional structures of hydrophilic polymers, which are able to imbibe large amounts of water. Typically, hydrogels do not dissolve in water solution thanks to different mechanisms such as crosslinking and chain entanglement. They are characterized by a number of physiochemical properties that make them suitable for a variety of biomedical applications. They can have natural origin, such as collagen, fibrin, gelatin or agarose, or they can be synthetic, such as polyacrylic acid and polyethylene oxide. Hydrogels present a series of properties that make them interesting for the fabrication of microfluidic devices mimicking an in vivo environment [85] such as: (i) some of them are inexpensive (e.g., agarose) and, in general, they are relatively easy to be used as a microfluidic devices fabrication material (if compared with other techniques, such as lithography or direct laser writing); (ii) optically clear over visible spectral range; (iii) permeable for small molecules, being in this way ideal as a selective diffusion barrier; (iv) depending on the choice of hydrogel, they can strongly promote or inhibit cell adhesion [86]; (v) their mechanical properties (such as their Young modulus) are comparable to the one of biological tissues and can be tuned depending on the fabrication procedure. From the fabrication point of view, they can be shaped exactly as PDMS using soft lithography procedures. Indeed, many hydrogels are liquid above certain temperature (thermogel) [86], so they can be poured on a mold and then peeled off once solidified. The obtained open-channels can be either closed in between two rigid slabs and sealed by pressure, or they can be attached to another hydrogel lid and fused together by partial melting of the contact interface [87]. Alternatively, they can be injected in existing microchannels (for instance, PDMS ones) and then solidified, acting as an embedded element in the fluidic network.

An example of the use of hydrogel (agarose)-based MCMA for the study of chemotaxis, even if without mechanical confinement, is presented by Cheng et al. [62]. The device presented three parallel channels. The chemoattractant is injected in the first one, while buffer solution is injected in the third. Thanks to the hydrogel permeability, a chemical gradient is established between the two non-communicating channels. This same chemical gradient is present also in the intermediate channel. In this way, it is possible to inject the biological sample in the central channel and observe the chemotaxis-induced migration, avoiding liquid media exchange between the various compartments of the device. Choi et al. [63] exploited the tunable mechanical properties of agarose to fabricate a gel confiner device with different stiffness, as shown in Figure 4a. This device consists of a hydrogel lid that is placed on the top of some 2D seeded cells in order to apply on them a known pressure and study their migration once confined. In particular, the agarose concentration was calibrated in order to mimic the mechanical stiffness corresponding to brain, lung, skin or spleen. Wang et al. [64] instead realized some straight channel array using collagen–alginate hydrogel using the replica molding approach (see Figure 4b). By changing the Ca${}^{2+}$ concentration in the solution, they were able to tune the stiffness of the structures in a range of 0.3–20 kPa, which is comparable to most biological tissues up to tumoral ones. They investigated the motion strategies of MDA-MB-231 breast cancer cells depending on the combination of channel width and stiffness and observed how this last parameter influences the transition from mesenchymal to ameboid migration mode.

A different approach is to embed the cells directly in the hydrogel, exactly as they would be in a biological tissue in a real case scenario, such as in the devices reported in Figure 4c,d. These type of devices are built in a modular way [88], combining three elements: an access channel for the cell seeding; a gel matrix region; and a means of controlling the experimental parameters (such as flux, pressure or chemical gradient). The gel region can be shaped depending on the requirement of the experiment. For instance, its width can be sized in order to guarantee a desired chemical gradient steepness across the two edges, or its height can be chosen in order to guarantee optical accessibility through the whole volume. A second important aspect of the fabrication process is how the hydrogel is introduced in the microfluidic device. For instance, the gel can be injected in its liquid phase in an empty channel in order to completely fill it, or it can be shaped using hydrodynamical focusing in order to confine it in a slab in the center of the channel [88]. These processes can apply a considerable pressure on the material. For this reason, they are not advisable if the sample under analysis (i.e., the cells) is already embedded in the hydrogel matrix. In this second case, the hydrogel can be instead deposited in the microchannel before the sealing.

The permeability of the hydrogel matrix can be used to study the effect of liquid media flux and pressure on the migrating cells. Huang et al. [65] embedded the cells (MDA-MB-231) in a collagen matrix pumped inside a microfluidic device, and they used the system to simulate interstitial flow inside a biological-like ECM. In this case, the confinement on the cells was operated by the hydrogel matrix itself that acts as ECM, and the microfluidic system was used in order to accurately control the experimental condition in a lab-on-a-chip fashion, simulating tissue perfusion. The matrix itself can be coated or functionalized in order to study the cell–ECM interaction. Anguiano et al. [66] embedded in a collagen–Matrigel matrix H1299 lung cancer cells using two parallel lateral channels to establish a chemical gradient. Before injecting the hydrogel, they marked with a fluorescent dye the collagen structures. In this way, they were able to characterize the cell motility in this biological-like environment and identify the adhesion points between cells and ECM, thanks to specific fluorescent labeling. Furthermore, they studied the interaction between the cells and the ECM, the collagen fiber displacement, and characterized the force exerted by the cells on the surrounding structure.

Truong et al. [68] developed a device with two different regions filled with hydrogel, which were in contact one with the other. In their experiment, the first gel region encapsulated SUM-159 breast cancer, while the second one was cell-free and acted as a migration area only. The hydrogel used to fill the first region was changed (Matrigel, collagen I and a mixture of the two) in order to study the effect of the ECM on the migrating cells, while the second region was filled with collagen only, acting as a stroma region. From their analysis, they showed that the mixed hydrogel can be used to reduce matrix disruptions during cell migration, allowing for long-term reliable experiments. As a second step, they exploited microfluidics environment control capability in order to stimulate the cells with a transient gradient of epidermal growth factor. In this way, they were able to characterize and compare the increased cell speed in all three dimension of the space, having at the same time a control subset of cell on the same platform. As a third step, the authors embedded in the secondary region cancer-associated fibroblasts (CAF) cells in order to characterize cell-to-cell interaction during the migration process. As a result, the presence of a secondary cell line increased cancer cell motility, which was possibly induced by a chemoactractant effect created by CAFs cells themselves.

Ayuso et al. [67] used a collagen-based hydrogel ECM to simulate the migration of C-6 glicoblastoma cells toward or away from a capillary clogged by a thrombosis, i.e., exploiting microfluidics to create a transitory stimulus. In this configuration, the intermediate hydrogel mimics the biological tissue, while the two parallel external microchannels act as parallel blood vessels, delivering nutrients to the cells for the whole duration of the experiment. By stopping the flow of one of the two channels, they simulated the thrombosis event, which characterized the migration of the cells toward the working microchannel and studied the change of the morphology of the population along the migration direction.

Hydrogel matrices were used in microfluidic platforms to study biological phenomena other than single cell migration, such as for instance angiogenesis. Trappmann et al. [69] developed a non-swelling hydrogel and finely tuned its properties, such as stiffness and crosslinking, in order to characterize the transition of endothelial cell invasion strategy from single-cell migration and the multicellular, strand-like invasion. In particular, they showed that the discriminant factor between the two behaviors is the degree of degradability of the hydrogel matrix.

In general, hydrogels are an interesting platform for the realization of microfluidic devices, thanks to their specific aforementioned chemical and mechanical properties. The use of hydrogel matrices to simulate biological ECM in migration assays is a promising application, as they can faithfully replicate interesting characteristics such as stiffness or perfusion of the tissues. At the same time, being transparent at visible wavelengths and with a low light scattering, they allow for direct optical inspection and cell morphology study. On the other hand, as the internal structure of these hydrogel matrixes is random, it is not possible to directly engineer the mechanical stimulus applied to the migrating cell. This aspect limits the analysis to a statistical observation of population characteristics, rather than probing the specific response of the cell to a given stimulus, as was performed instead in the case of hydrogel-based microchannels.

## 4. Microfluidic Devices for Migration Assays Realized by Femtosecond Laser Micromachining

Femtosecond laser micromachining (FLM) of microfluidic devices for cell assays allows fabricating biochips with complex 3D geometries in several materials, as glass or polymer, with micrometer (and even nanometer) accuracy, and with a robustness that is difficult to obtain with PDMS and hydrogels strategies.

FLM is implemented by focusing a femtosecond laser beam inside a material that is transparent to the laser wavelength. Due to the high intensities reached at the focal spot, non-linear absorption phenomena take place, leading to a permanent modification of the material properties. In the case of glasses, under specific irradiation conditions, it will lead to a local increase in the etching rate that will allow obtaining embedded microfluidic channels after a chemical etching process. This regime is exploited to fabricate 3D microfluidic devices, inside glasses such as fused silica or Foturan [89,90] with no need for a clean room facility or a post gluing/bonding process to close the device. The technique is known as Femtosecond Laser Irradiation followed by Chemical Etching (FLICE).

In the case of photopolymers, it will induce a local photo-crosslinking of the material and consequent polymerization of sub-micrometer features [91], which is also called two-photon polymerization (TPP), thus allowing the fabrication of 3D structures with arbitrary geometries. These structures can be used to modify the topography of migration surfaces [92] or even be fabricated inside complicated microfluidic devices.

There are a few examples in the literature that exploit FLM for the development of microfluidic devices to study cell motility under a constrained environment. The first applications were dedicated to the fabrication of free-standing scaffold-like structures fabricated by TPP in order to study human fibrosarcoma (cell line HT1080) migration [72]. Tayalia et al. [72] manufactured scaffolds with pore sizes ranging from single cell size to ten times larger (from 12 to 110 µm). This 3D structure allowed demonstrating that the migration speed through these structures is faster than in the 2D migration experiments, thus paving the way to more systematic studies of cell migration in 3D environments.

In order to have a more controllable chemical environment for the cell growth, Olsen et al. [73] fabricated similar structures (woodpiles-like) inside a commercial plastic microfluidic chip designed for chemotaxis experiments, as shown in Figure 5a. They applied this device to study dendritic cells migration under a CCL21 chemoattractant gradient. To provide a better biological environment, the structure was filled with collagen in a concentration that still permitted establishing a chemoattractant gradient. They observed that dendritic cells were able to migrate through 8 × 8 μm${}^{2}$ pores, while migration in woodpile structures with lower porosity was rarely seen. A more complicated configuration was presented some years after by the same group [93]. They fabricated linear channel constructs with profiles ranging from 10 × 10 μm${}^{2}$ to 20 × 20 μm${}^{2}$, which allowed a more precise control of the CCL21 gradient. They demonstrated that under these constrained conditions, the chemoattractant gradient steepness governs the directed migration speed rather than the channel size.

A possibility is to take advantage of FLM versatility to fabricate both the microfluidic chip and the migrating assay constrains. Indeed, Sima and co-workers developed a fully FLM-fabricated device to study the migration potential in sub-micrometer constrictions of PC3 cancer cells under a chemical gradient [70]. The microfluidic device was fabricated in Foturan glass by the FLICE approach with a subsequent thermal treatment to restore the glass optical transparency to guarantee clear imaging of cell migration. Afterwards, TPP was used to include a series of small sub-channels inside the microfluidic chip with a rectangular profile presenting lateral open widths ranging from 2 to 0.7 μm (see Figure 5b). These interesting sub-micrometer features were possible thanks to the unique capabilities of TPP structuring. A different approach was presented later on by fabricating the whole device in glass using the FLICE technique [71]. They created long pillars inside the microchannel that, after the thermal treatment, deformed into nanochannels narrower than 1 μm with a height of 6.75 μm and a length of over 50 μm.

Additional exploitation of the FLM 3D capabilities was demonstrated by Ficorella and co-workers [29,74] in the fabrication of a lab-on-a-chip to study cell migration under micrometer size confinement, in channel constructs with different geometries as the ones reported in Figure 5d–f. The devices were fabricated with a hybrid approach: FLICE was first used for the fabrication of cell growth chambers, and afterwards, TPP was employed for the realization of the channel-like constrictions and for the sealing of the microfluidic device. In addition to the micrometer size of the constraints and the tailoring of their geometry, this approach allows for a high-resolution imaging of the migrating cells thanks to the thin and transparent device floor. This micro-constriction configuration has been used for more systematic biological studies such as the adaptability of different cancer cell lines (MCF-10, MDA-MB-231) to micro-constrained environments and to study different migration behaviors depending on cancer cell invasiveness [29,30].

Although FLM fabrication is more expensive and time consuming than PDMS and hydrogels soft lithography, the possibility of reusing the devices makes FLM an attractive alternative. In fact, the non-porosity of the materials and the robustness of the fabricated devices allows cleaning them with solvents such as ethanol or detergents such as Mucasol [74]. Moreover, the potential of this technique to modify several different materials with even sub-micrometer accuracy and its inherent 3D capabilities will allow including new functionalities that might improve in vitro cell-migration assays. We envisage that thanks to the aforementioned characteristics, it will have an important role in the prototyping of future microfluidic devices for cell migration studies.

## 5. Conclusions

Microfluidic platforms are emerging as a valuable tool for the study of cell migration under spatial confinement. Although their fabrication requires specific competencies and a careful design and their use can be less intuitive and straightforward than the more traditional methods, such as Boyden chambers or scratch assays, these microfluidic devices offer several advantages, such as a precise control of the biological and chemical environment, an easy interface, and the possibility of being tailored to the requirements of the specific biological experiment. Indeed, MCMA platforms constitute a powerful tool to perform more focused cell migration experiments and improve the quantity and the quality of information that can be extracted from these biological studies if compared to traditional approaches.

Thanks to the precise control on the geometry and on the characteristics of the MCMA, it is possible to perform several different experiments, choosing the type of ECM and the mechanical stimulus or the environmental conditions. Microfluidics enables the realization of more complicated assays, opening the possibility to inquire into aspects of cell migration that would not be detectable with standard techniques. Furthermore, we envisage that in the near future, the possibility to integrate them with embedded sensors, or new microscopy techniques, will improve the interpretation of cell behavior.

The choice of the fabrication technology employed in the MCMA realization has some consequences on the possible characteristics of the final device. PDMS and hydrogels are inexpensive and easy to handle materials that can be used to straightforwardly realize microfluidic prototypes. The process of soft lithography allows the realization of minimum feature sizes comparable to the one of the single cells (few micrometers); thus, they can be used for the realization of MCMA to study spatial confinement. At the same time, complex geometries can be realized to investigate specific aspects, such as the memory effect of a series of constrictions or the traction force that the cells exert on bendable pillars.

Still, this type of technology remains intrinsically planar, and the shape of the single channel or constriction is dependent on the characteristics of the mold; thus, it cannot be easily changed. Additionally, it is hard to push the geometrical limit toward or below the micrometer level. Furthermore, these materials tend to adsorb proteins with time, and this strongly limits their use in the field of pharmacology and drug-related studies [94].

Laser-writing techniques such as FLICE and TPP offer the possibility to realize custom-made geometries with micrometer or even sub-micrometer precision and the capacity of shaping the microchannels in a three-dimensional way. This allows the realization of structures for the analysis of cell migration under mechanical confinement with features hardly reproducible with other techniques. The materials that have been used for the laser fabrication of MCMA platforms are glass and polymers, which are known to be rigid and non-porous, thus avoiding the absorption of proteins of other biological materials. Moreover, they are proved to be biocompatible and mostly inert, thus allowing chemical and biochemical analysis.

Although the realization of these devices by laser-writing techniques requires dedicated equipment, which limits the fabrication of MCMA with this approach to specialized laboratories, the devices fabricated by FLM offer the advantage of being reusable several times. This technology expresses its advantages in the prototyping and in the realization of customized devices, with strict requirements from the point of view of the design complexity. Furthermore, it allows the integration of other possible technological elements, such as micromechanical or optical components, in the same platform and fabricated with the same setup, pushing the devices from simple, single-task systems toward multi-analysis lab-on-a-chip platforms [89].

## Figures and Tables

**Figure 1 biosensors-12-00604-f001:**
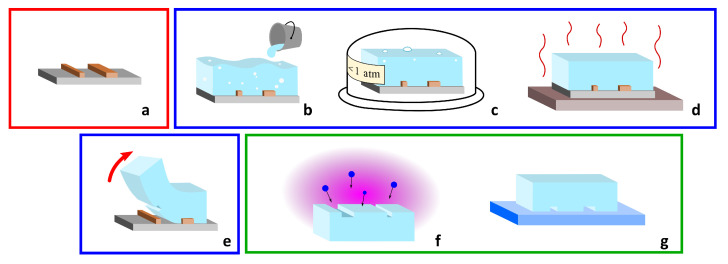
Procedure for PDMS microfluidic device realization: first, the negative mold of the desired fluidic network is realized using, for instance, lithographic techniques (**a**); as a second step, the network is replicated on the PDMS: the liquid elastomer is poured on the mold (**b**), then, it is degassed under a low vacuum chamber (**c**), it is cured on a hotplate (**d**), and finally, it is peeled off the mold (**e**); last, the PDMS is bonded on a sealing substrate, for instance, glass: the surface is activated using a plasma treatment (**f**), and then, the whole devices is permanently assembled (**g**).

**Figure 2 biosensors-12-00604-f002:**
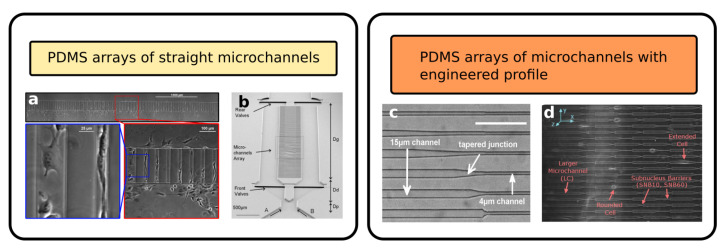
Examples of cellular migration assay realized with arrays of straight microchannels in PDMS: (**a**) array of microchannel connecting culture and chemoactractant chambers. Each panel is a close-up of the previous, showing the detail of the array and the shape of the cells in two channels with two different widths (Reprinted with permission from Ref. [49] Copyright 2012 Tong et al.); (**b**) advanced geometry integrating several components in the same device, such as: an array of straight channels (center), a connection channel to balance the chemical gradient (bottom) and two microfluidic valves (top and bottom) (Reprinted with permission from Ref. [50] Copyright 2007, The Royal Society of Chemistry); (**c**) variable cross-section channels with different tapering angles (Reprinted with permission from Ref. [53] Copyright 2011, Mak et al.); (**d**) array of microchannels with periodically modulated width (Reprinted with permission from Ref. [54] Copyright 2013, The Royal Society of Chemistry).

**Figure 3 biosensors-12-00604-f003:**
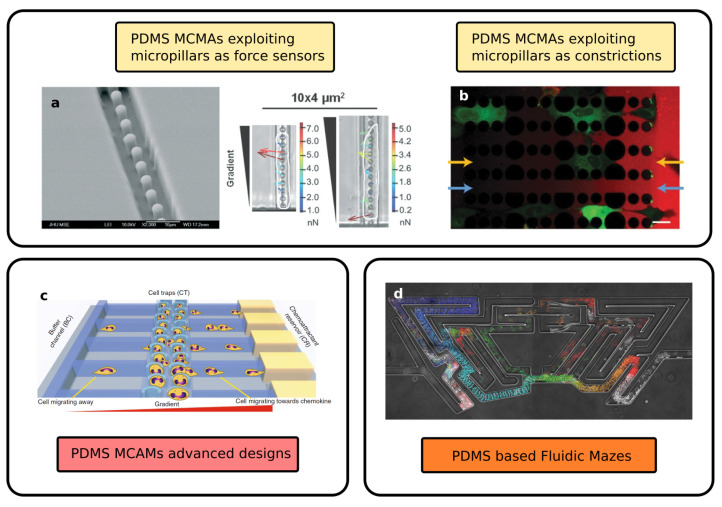
Advanced configurations of PDMS microfluidic assays: (**a**) PDMS pillar structures combined with constricting channels, used to study cell adhesion forces. The second panel shows how the displacement of the pillars is related to the force applied by the migrating cell (contoured in white) (Reprinted with permission from Ref. [56] Copyright 2013, The Royal Society of Chemistry); (**b**) Combination of microchannel and pillar structures (colored in black), used to study nuclear rupture during cell migration through high confining constriction. The cells are reported in green, while the chemoactractant is reported in red. The arrows highlight the constricting channel and the connection channel used for chemoactractant diffusion in orange and blue, respectively (Reprinted with permission from Ref. [59] Copyright 2015, The Royal Society of Chemistry); (**c**) PDMS chip containing cell traps (in the center) and a double set of constriction channels for bidirectional chemotaxis study (Reprinted with permission from Ref. [57] Copyright 2014, Nature Publishing Group); (**d**) Fluidic maze used to investigate cellular migration decisions. The different time-steps are represented in different fake colors (Reprinted with permission from Ref. [60] Copyright 2020, AAAS).

**Figure 4 biosensors-12-00604-f004:**
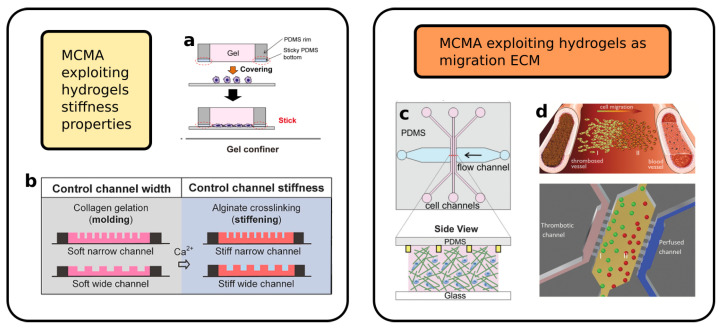
Examples of microfluidic migration assays realized with hydrogels:(**a**) Hydrogel lid used to apply controlled mechanical load on migrating cells (Reprinted with permission from Ref. [63] Copyright 2021, Choi et al.); (**b**) Microchannels array with different size and stiffness realized exploiting hydrogel mechanical–chemical properties (Reprinted with permission from Ref. [64]. Copyright 2019 American Chemical Society); (**c**,**d**) Hydrogel-based matrix used to study chemotaxis (Reprinted with permission from Ref. [67] Copyright 2017, Ayuso et al., and Ref. [66] Copyright 2020, Anguiano et al.).

**Figure 5 biosensors-12-00604-f005:**
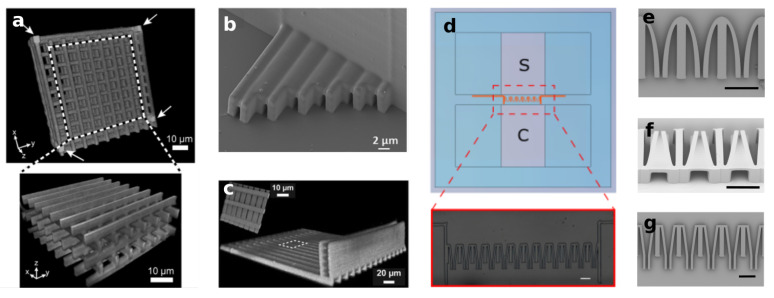
Examples of polymeric structures for cell migration assays realized with TPP: (**a**) Confocal image of woodpile structure, integrated inside a microfluidic channel (the microchannel extends along x axis) (Reprinted with permission from Ref. [73]. Copyright 2013, The Royal Society of Chemistry); (**b**,**c**) Arrays of straight microchannels realized inside a glass and plastic microchannels (Reprinted with permission from Ref. [70], Copyright 2019 American Chemical Society and from Ref. [93], Copyright 2015 Springer Science+Business Media New York); (**d**) Example of glass–polymer microfluidic cell migration assay, completely realized with femtosecond laser micromachining (scalebar 100 µm); SEM images of different possible micro-constriction geometries, realized with TPP, with (**e**) Elliptical profile, (**f**) Three-dimensional profile and (**g**) Combination of funnel profile and straight channels (scalebar 100 µm, Reprinted with permission from Ref. [74]. Copyright 2021, Sala et al.).

**Table 1 biosensors-12-00604-t001:** Overview of the possible microfluidic cell migration assays approaches.

References	Details	Constriction Characteristics	Schematic
	**PDMS—Straight channels array**		
Rolli et al. [47] ^1^ Fu et al. [48] ^1^ Tong et al. [49] ^2^ Irimia et al. [50] ^1^ Spuul et al. [51] ^1^ Zhou et al. [52] ^2^ Mishra and Vazquez [43]	Chemotaxis analysis, comparison of migration behaviors depending on channels dimensions or chemical stimuli	Constant cross-section. Channels characteristic dimension from 50 to 3μm	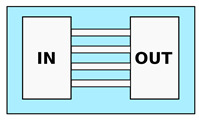
	**PDMS—Microchannels with engineered profile**		
Mak et al. [53] Mak et al. [54] Ma et al. [55] Raman et al. [56] ^2^ Boneschansker et al. [57]	Study of migration strategies depending on the local 3D channel geometry, such as tapering or height modulation; Integration of cell traps or bendable micropillars as cell force probes	Variable cross-section. Width varying form 50 to 4μm. Height varying from 15 to 10μm	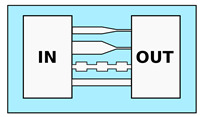
	**PDMS—Micropillars**		
Doolin and Stroka [58] ^2,3^ Davidson et al. [59] ^1^	Use of pillar arrays as ECM; Analysis of 2D cell motility depending on environment geometry; Study of cell migration through sub-nuclear dimension pores	Variable cross-section and 2D profile. Width varying from 50 to 2μm	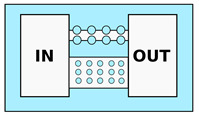
	**PDMS—Fluidic Maze**		
Tweedy et al. [60] Belotti et al. [61]	Study of cell decision making during migration and cellular environment probing capacity (e.g., fluidic resistance or self-induced chemical gradient)	Constant single channel cross-section. Bifurcations, corners and widenings. Channels dimension from 5 to 3μm	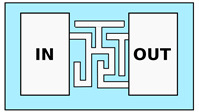
	**Hydrogels—Microchannels**		
Cheng et al. [62] ^1^ Choi et al. [63] ^1^ Wang et al. [64]	Chemotaxis analysis, comparison of migration behaviors depending on channels dimensions or chemical stimuli; Possibility to modify mechanical properties of the channels, such as their stiffness	Constant cross-section. Channels dimension from 14 to 3.5μm	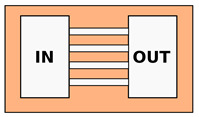
	**Hydrogels—Migration matrix**		
Huang et al. [65] Anguiano et al. [66] Ayuso et al. [67] Truong et al. [68] Trappmann et al. [69] ^4^	Use of hydrogel matrix as ECM, mimicking biological tissues in terms of porosity and stiffness. Possibility to embed the cells directly inside the matrix	No opened channels, cells migrate through the hydrogel. Possible presence of voids or pores with micrometric dimension. Mechanical stiffness ranges from few tens of Pa to tens of kPa (e.g., 18 kPa)	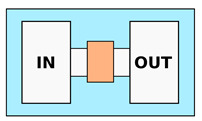
	**FLM—Glass-based devices**		
Sima et al. [70] ^2^ Sima et al. [71] ^2^	Microchannels with arbitrary cross-section realized in the bulk glass substrate.	Variable cross-section. Width varying from 5 to 0.9μm	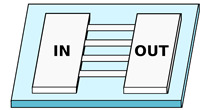
	**FLM—Two-photon polymerization devices**		
Tayalia et al. [72] Olsen et al. [73] ^2^ Ficorella et al. [29] ^1,2,3,4^ Sala et al. [74] ^1,2,3,4^	Polymeric 3D structures working as micrometer spatial constrains fabricated inside wider microfluidic channels. Possibility to arbitrary adjust the target geometry, from scaffolds or woodpiles to microchannels with arbitrary cross-section	Scaffold-like structure with porous size from 5×5μm${}^{2}$ to 15×15μm${}^{2}$. Channel with variable cross-section, from 20×20μm${}^{2}$ to 5×5μm${}^{2}$	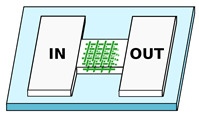

Devices coated with: ^1^ fibronectin; ^2^ collagen; ^3^ Pluronic F127; ^4^ poly-d-lysine.

## Data Availability

Data sharing not applicable.

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
