# Peer review of "Microfluidic Lab-on-a-Chip for Studies of Cell Migration under Spatial Confinement"

_biosensors, 2022, doi:10.3390/bios12080604_

Round 1

Reviewer 2 Report

The manuscript entitled "Study of cell migration under mechanical confinement in microfluidic lab-on-a-chip " by Sala et al. showed how many different kinds of microfluidic devices have been used in the field to study cell migration under confinement. The authors have discussed three different kinds of microfluidic devices: PDMS-based, Hydrogels-based, and femtosecond laser micromachining. Although the authors meticulously documented the different variances of the microfluidic devices, the review is monotonous with a plethora of discrete information and little linkage, which does not create enough excitement/understanding/curiosity for the readers. The review needs significant improvement in the way different parts are presented right now. These can be some points that can help the authors to improve their manuscript.  

1.    It is always best to use more schematic figures in a review paper to convey the general message and keep the reader more engaged. Please use an introductory schematic figure highlighting these points (can include additional features)

a.       The general setup of microfluidic devices (different compartments) and how they work

b.       How many different kinds of biological questions can be addressed using microfluidic devices.

c.       How the data interpretations are made to answer specific biological questions.

2.    In the title, the authors highlighted "study of cell migration...", but in most parts, the authors described what different kinds of devices had been used but did not focus very clearly on characterizing specific features of the cell migration and how an improved device helped in solving the biological problems. For example, in several places, authors have mentioned what kinds of cell lines were used in the study but did not focus

a.       if there are inherent differences in the cell migration of different cell lines,

b.       if there are differences in cell migration between healthy cells vs. disease cell line

       Please add more importance to the biological questions or the problems which is being addressed by the devices.

3.    The authors mentioned how several improvements had been made to the existing devices but it was not mentioned clearly the drawback of the earlier models and where they were not good enough to address specific biological questions. How is the newly designed model able to solve the problem?

4.    As the paper is jammed with a vast amount of information, it is better to use tables to combine

a.        different microfluidics devices used,

b.       specific limitation of each device,

c.       how the new device is better than the earlier one

d.       the improvements help to address specific biological questions (but not just in what cell line it was used)

5.    This review provides very few original thoughts from the authors. The authors need to make make a separate subsection of future direction, where the authors can discuss

a.       What can be the different regions of the biological field where microfluidic devices can be used in the future?

b.       How can the present devices be improved by using new technology or modifying the existing devices' design?

c.       Limitations in the present research

6.    To reduce the monotonous tone of the paper, the authors need to make more subsections.

7.    Many of the figures need to be improved. Just cutting and pasting device pictures from published papers won't be good enough. In a review paper, the figures should not be presented in such a way that the readers need to go back to the original paper to understand what is being presented. The authors need to add more context to the figures by either highlighting specific information in the present figures or along with the images they have taken from other papers; they should provide some schematic to highlight the main takes aways. For example,

a.       Figures 2a and 2c are not clearly visible. If the authors could not find better representative images of the devices, then use schematics to draw/highlight specific information they want to highlight.

b.       In figures 3a and 3b, very little can be understood.

8.    The authors should be more specific about the information they provide; for example, "Anguiano et al. [58] embedded in a collagen-Matrigel matrix some lung cancer cells using two parallel lateral channels to establish a chemical gradient." Please replace 'some' lung cancer cells with the specific cells used.

9.    Please use the figure number in the figure's top-left panel rather than the bottom right.

10.  The authors have added a subsection Patents (page 12, line 485) but did not provide any details

11.  The authors should make a concluding figure to summarize all different devices and their specific properties, advantage and disadvantages.

12.  As this paper is submitted to the Biosensor journal, the authors should try to include some references from the journals published in the Biosensor Journal. 

Round 2

Reviewer 1 Report

Most of the previous comments have been considered in the new version of the manuscript, and I totally understand the scope of the review now, focusing on the design and fabrication of systems.

English style and spelling has to be checked carefully on the new text parts.

Reviewer 2 Report

The authors have nicely addressed most of the significant concerns. This article can be published with no further change.